

# Rogue Wave Indicators from Global Models and Buoy Data

Laura Azevedo[1], Gabriel Marcon[2], Steven Meyers[1], Mark Luther[1]

[1]College of Marine Science, University of South Florida, St. Petersburg, Florida, 33701, USA
[2]University of the People, Pasadena, California, 91101, USA

*Correspondence to*: Laura Azevedo (laurac8@usf.edu)

**Abstract.** Rogue waves pose substantial risks to maritime operations and offshore infrastructure, yet their formation mechanisms and predictability remain poorly understood. This study analyses real rogue wave occurrences using in situ observations from CDIP wave buoys from the Filtered Ocean Wave Data (FOWD) dataset and model-based estimates from ERA5 reanalysis and the ECMWF CY47R1 high-resolution hindcast. Seasonal distributions, wave height comparisons, and

spectral analyses reveal that models systematically underestimate extreme wave events due to spectral smoothing and spatial averaging. A key finding is that rogue waves are usually preceded by a sharp decrease in crest-trough correlation below 0.5, followed by a rapid increase above 0.6, indicating a transition to a more structured wave field. This pattern, accompanied by spectral bandwidth narrowing and increased relative energy in the 0.25–1.5 Hz range, suggests energy focusing mechanisms play a critical role. Analysis of rogue wave events at four CDIP buoy stations show that the crest-trough correlation

parameter alone is not a good rogue wave indicator, but its temporal variability is. These results highlight the need for improved modelling by integrating dynamic wave field specific parameters and high-resolution numerical models to enhance rogue wave risk assessments on a global scale.

## 1 Introduction

Rogue waves, also known as freak waves, represent one of the most enigmatic and dangerous phenomena in oceanography.

These waves, characterized by their exceptional height relative to the surrounding sea state, pose significant threats to maritime safety, including risks to vessels, offshore infrastructure, and coastal regions (Bitner-Gregersen, 2015; Didenkulova, 2020; Didenkulova et al., 2023; Kharif et al., 2009). Defined as waves with a maximum height (Hmax) exceeding twice the significant wave height (Hs), rogue waves have been documented across diverse oceanic environments, from deep waters to nearshore zones (Dysthe et al., 2008; Toffoli et al., 2005). The global significance of rogue waves was

underscored by the observation of the New Year's Wave at the Draupner oil rig in 1995, which triggered a renaissance in their scientific study (Adcock et al., 2011; Haver, 2003).

Over the past two decades, various physical mechanisms have been proposed to explain the occurrence of rogue waves. These include nonlinear processes such as modulation instability (Benjamin-Feir instability), wave focusing through

dispersion, and interactions with bathymetric features (Cattrell et al., 2018; Fedele et al., 2016; O. Gramstad et al., 2018).



Analytical models, numerical simulations, and laboratory experiments have provided significant insights into these mechanisms(Onorato et al., 2005; Onorato et al., 2003; Toffoli et al., 2011). However, the real-world application of these theories remains limited due to the rarity of rogue wave events and the sparse availability of in-situ observations (Azevedo et al., 2022; Barbariol, Benetazzo, et al., 2019; Baschek & Imai, 2011). This scarcity of high-quality observational data,
particularly from buoys, has constrained the development of robust predictive models (Orzech & Wang, 2020). From a scientific perspective, rogue waves challenge conventional understanding of ocean dynamics.

Recent advancements in reanalysis datasets and wave modeling offer a promising avenue for addressing these limitations. Reanalysis products such as ERA5 and higher-resolution models like ECMWF's CY47R1 provide global coverage of wave
parameters, integrating observational and modeled data to simulate historical ocean conditions. While ERA5 assimilates observational data and is designed to reflect real-world conditions, its spatial resolution (approximately 40 km) can result in the smoothing of extremes (Barbariol, Bidlot, et al., 2019; ECMWF, 2021; Hersbach et al., 2020). In contrast, the CY47R1 wave model, with its higher resolution (18 km) and enhanced spectral fidelity, is potentially more sensitive to localized, transient events such as rogue waves, albeit without assimilating observational wave data (Cavaleri et al., 2022; ECMWF,
2022; Lobeto et al., 2024).

While traditional theories of formation have emphasized nonlinear modulational instability (Onorato et al., 2009; Teutsch & Weisse, 2023), recent studies highlight the role of spectral energy distribution, wave group dynamics, and crest-trough correlation in governing rogue wave emergence (Cicon et al., 2024; Gemmrich & Cicon, 2022; Gemmrich & Thomson,
2017; Hafner et al., 2021b). Given the severe implications of these waves for navigation, offshore energy production, and climate-driven wave climate shifts, there is a growing need for improved methodologies that detect and predict rogue wave conditions using this new knowledge.

Numerous studies have investigated the mechanisms underlying rogue wave formation (Babanin & Rogers, 2014; Bennett et
al., 2012; Gemmrich & Cicon, 2022; Onorato et al., 2009). Early research primarily focused on the role of modulational instability, which predicts that under specific conditions, wave groups can undergo nonlinear focusing, leading to extreme wave growth (Adcock et al., 2015; Donelan & Magnusson, 2005; Horikawa & Maruo, 1987; Janssen, 2003). However, observational evidence suggests that rogue waves also frequently occur in environments where modulational instability is weak or absent, particularly in wind-sea dominated conditions (Fedele et al., 2016; Hafner et al., 2021b). Recent
advancements in spectral wave analysis have pointed toward wave group dynamics, spectral bandwidth narrowing, and crest-trough correlation as dominant factors contributing to rogue wave development. Studies utilizing high-resolution buoy measurements, such as those by Gemmrich and Cicon (2022) and Häfner et al. (2021), have shown that narrower spectral bandwidths and increased energy concentration in specific frequency ranges lead to increased wave focusing (higher crest-trough correlation), thereby enhancing the probability of rogue wave formation.




Despite these advances, a major limitation in rogue wave research is the reliance on localized in situ measurements, which are sparsely distributed and primarily concentrated in coastal waters (Barbariol, Benetazzo, et al., 2019; Baschek & Imai, 2011). Global wave modelling and reanalysis datasets, such as ERA5 reanalysis and the ECMWF CY47R1 high-resolution hindcast, remain the most used tools for vessel design criteria, routing configuration, and operational wave forecasting.

These models integrate spectral wave dynamics and physics-based simulations to reconstruct past wave climates and provide probabilistic wave hazard assessments  (ECMWF, 2020, 2021, 2022). However, evidence suggests that these models systematically underestimate extreme wave occurrences, particularly rogue waves, due to their reliance on smoothed spectral representations and linear parameterizations (Barbariol, Bidlot, et al., 2019; Campos et al., 2018; Janssen, 2015). This discrepancy presents significant challenges for maritime safety and operational oceanography, as rogue waves continue to be

an unpredictable and underrepresented threat in global wave forecasts.

Building on these developments, this study aims to bridge the gap between observational and modelled perspectives of rogue wave events. By comparing FOWD-derived buoy data (Hafner et al., 2021a) with the ERA5 reanalysis (ECMWF, 2021) and ECMWF CY47R1 hindcast wave model (ECMWF, 2022), we seek to identify patterns or indicators in the modelled data that

correspond to rogue wave occurrences observed in situ. The overarching goal is to develop methodologies for mapping rogue wave probabilities on a global scale, leveraging the extensive spatial and temporal coverage of reanalysis and high-resolution wave models.

A key component of this study is the assessment of seasonal rogue wave distributions, highlighting the climatological

dependence of rogue wave occurrences (Cattrell et al., 2019; Jonathan & Ewans, 2011). Statistical comparisons of Hmax and Hs distributions, coupled with density scatter plots and spectral analyses, provide insight into the discrepancies between FOWD observations and modelled outputs. Additionally, spectral parameters such as wave skewness, kurtosis, and the Benjamin-Feir Index (BFI), which are available in these datasets, are investigated as indicators of nonlinear wave interactions, wave group coherence, and instability mechanisms (Azevedo et al., 2022). While these spectral metrics have

been widely employed to describe rogue wave likelihood, their ability to capture real-world extreme wave events remains unproved when applied to large-scale numerical models (Lobeto et al., 2024).

We advance to evaluate the effectiveness of crest-trough correlation as a more promising indicator of rogue wave formation (Cicon et al., 2024; Gemmrich & Cicon, 2022; Hafner et al., 2021b; Teutsch & Weisse, 2023). Recent research suggests that

rogue waves are better sustained in sea states where crest-trough correlation exceeds 0.6 (Hafner, 2022), yet we hypothesize that this parameter alone should not be used as a reliable predictor, because it is very usual for waves to reach this 0.6 crest-trough correlation value. Instead, this present study hypothesizes that the dynamic evolution of crest-trough correlation - specifically, a sharp decrease, usually reaching below 0.5, followed by a rapid increase reaching above 0.6 - precedes rogue



wave events and may serve as an early warning sign of extreme wave amplification. To validate this hypothesis, detailed
case studies of rogue wave occurrences at four strategically located CDIP buoy stations—Station 098 (Mokapu Point, HI),
Station 154 (Block Island, RI), Station 162 (Clatsop Spit, OR), and Station 166 (Ocean Station Papa)—are conducted. These
analyses assess the temporal evolution of crest-trough correlation alongside spectral bandwidth narrowing and relative
energy concentration in the 0.25–1.5 Hz range, new possible significant parameters that influence rogue wave development
through energy redistribution and wave group dynamics (Gemmrich & Thomson, 2017; Saulnier et al., 2011).


The significance of this research extends beyond theoretical advancements in rogue wave physics. Given the sparse global
coverage of in situ wave measurements, primarily concentrated along the coastlines of North America and Europe, there is
an urgent need to extrapolate rogue wave risk assessments beyond these regions. The underrepresentation of rogue waves in
global wave models poses a substantial risk to shipping operations, offshore platforms, and coastal infrastructure, where
extreme waves can cause structural damage and human casualties (Bitner-Gregersen & Toffoli, 2013; Bitner-Gregersen &
Gramstad, 2015; Bitner-Gregersen et al., 2018; Odin Gramstad et al., 2018). By addressing these challenges, this study aims
to contribute to the advancement of operational oceanography and the development of predictive tools for extreme wave
events. Ultimately, the findings presented here provide a foundation for more accurate rogue wave risk assessments and
support the implementation of enhanced forecasting methodologies for extreme sea states worldwide using models (Cicon et
al., 2024; Cicon et al., 2023; Orzech, 2019).

## 2 Methodology

The methodology employed integrates three different data sources to examine the measurement and perception of rogue
waves, utilizing in situ observations from CDIP buoys – from the FOWD dataset, ERA5 reanalysis data, and high-resolution
hindcast data from ECMWF's CY47R1 wave model.


The buoy data used were obtained from the FOWD (Filtered Ocean Wave Data) dataset curated by Dion Häfner, which
mainly filters wave with spectral significant wave height ($Hm_0 = \sqrt{4 \times m_0}$ , where $m_0$ is the zeroth spectral moment of the
wave energy spectrum) above 1m. The FOWD dataset is built from the Coastal Data Information Program (CDIP) buoy
network, primarily consisting of Datawell Directional Waverider buoys deployed across various regions around the United
States territories coasts, with data spanning several decades (Hafner et al., 2021a). FOWD processes raw buoy observations
through a running time window approach that accounts for the non-stationary nature of the ocean. It applies extensive quality
control measures, including spectral density estimation using Welch's method, crest-trough correlation analysis, and spectral
partitioning. The dataset includes over 4 billion wave observations, providing a high-resolution view of wave dynamics
(Hafner et al., 2021a). Key parameters extracted from the buoy data include maximum individual wave height, significant
wave height, Benjamin-Feir Index, wave spectral peakedness, spectral kurtosis, spectral skewness, spectral bandwidth



narrowness, dominant wave period, dominant directional spread, relative energy in the frequency range of 0.25 to 1.5 Hz, and crest-trough correlation. These parameters were selected based on their relevance to wave dynamics and their ability to characterize extreme events (Hafner et al., 2021b).

To complement the in-situ observations, reanalysis data from ERA5 were employed. ERA5, produced by the European Centre for Medium-Range Weather Forecasts (ECMWF), provides hourly estimates of atmospheric, land, and oceanic climate variables, including wave properties (ECMWF, 2021; Hersbach et al., 2020). The wave component of ERA5 is based on the third-generation WAM model, which simulates wave generation, growth, and dissipation (Liu et al., 2022). ERA5 assimilates observational data from buoys, satellites, and ships to produce a globally consistent dataset with a spatial

resolution of approximately 40 km and a temporal resolution of one hour (ECMWF, 2021). The wave spectra in ERA5 are discretized into 30 frequency bands and 24 directional bins, which may affect the representation of high-energy, short-lived wave events (Barbariol, Bidlot, et al., 2019; Janssen, 2015).

To address the limitations of ERA5, we included the ECMWF high-resolution hindcast dataset based on the CY47R1 wave

model. The CY47R1 dataset features a higher spatial resolution of approximately 18 km and a finer spectral resolution with 36 frequency bands and 36 directional bins (ECMWF, 2022). Unlike ERA5, CY47R1 does not assimilate observational wave data, relying solely on numerical simulations forced by ERA5 wind fields. Its higher resolution, however, tends to resolve extreme wave events more effectively, capturing transient rogue waves that might be smoothed out in ERA5 (Barbariol, Bidlot, et al., 2019; Lobeto et al., 2024). The CY47R1 hindcast from 2015 to 2021 was specially selected for this

study because it is the only global wave hindcast that was re-ran to include calculations of the wave parameters Dion Hafner found to be important for rogue wave identification, such as: crest-trough correlation, spectral bandwidth narrowness and relative energy in the frequency range of 0.25 to 1.5 Hz, which are the parameters this study means to analyze and test.

For all datasets, careful preprocessing was conducted to ensure consistency and comparability. The reanalysis and hindcast

data were chosen to the locations of the selected buoys using the nearest grid point method. Spatial and temporal matching were used to align model estimates with buoy observations. Additionally, wave parameters corresponding to those calculated from the buoys' records (maximum wave height, spectral skewness, spectral kurtosis, Benjamin-Feir Index) were extracted from the ERA5 and ECMWF hindcast datasets. Quality control measures were implemented to remove data gaps, filter outliers – making sure we were not filtering any rogue wave and ensure coherence across datasets (Hafner et al., 2021a).


Rogue waves were identified using established criteria, primarily based on the threshold condition Hmax is larger than 2 times Hs, where Hmax is the maximum wave height and Hs is the significant wave height (Dysthe et al., 2008; Fedele, 2016; Garrett & Gemmrich, 2008). This definition is commonly used in the literature although it does not capture the dangerousness of these waves since there is no wave height threshold, meaning a one-meter wave can be considered a rogue



wave if the significant wave height (average sea conditions) is lower than 0.5m (Fedele et al., 2016; Heller, 2006; Stansell, 2004; Wolfram et al., 2001). The smaller rogue wave present in the FOWD dataset had 2m of height since it filtered out waves with significant wave height below 1m. The identification and classification of rogue waves were conducted systematically across all datasets, ensuring consistency in comparisons.

The first phase of the analysis focused on comparing Hmax (maximum wave height) and Hs (significant wave height) distributions from FOWD, ERA5, and ECMWF CY47R1, identifying discrepancies in how rogue waves are represented across these datasets. Seasonal variability was investigated using whisker plots and maps, assessing how wave height extremes fluctuate throughout the year and whether numerical models effectively capture rogue wave probability under different oceanic conditions. Additionally, probability density functions including information on skewness, kurtosis, and the
Benjamin-Feir Index (BFI) are examined to assess the presence of non-Gaussian statistical behaviors and modulational instability signatures in the datasets (Bitner-Gregersen & Toffoli, 2012; Mori & Janssen, 2006; Nederkoorn & Seyffert, 2022; Orzech & Wang, 2020).

The second phase of the analysis focused on investigating specific rogue wave events and looking at the parameters which
lately have been consider more significant for rogue wave identification: narrowness, relative energy in the 0.25–1.5 Hz frequency range and crest-trough correlation. The spectral bandwidth narrowness parameter describes the concentration of wave energy within a specific frequency range, where a lower bandwidth implies a more focused wave spectrum with less energy dispersion across multiple frequencies. Narrower bandwidths facilitate the focusing of wave energy and increase the probability of constructive interference (Saulnier et al., 2011). This phenomenon is particularly important in conditions that
favor rogue wave generation, as a more concentrated spectrum enhances wave coherence and promotes nonlinear wave interactions (Gemmrich & Thomson, 2017).

The relative energy within the 0.25–1.5 Hz frequency range serves as an indicator of the balance between long-period swell waves and shorter wind-sea components. When a significant portion of wave energy is contained within this range, long-
period swells dominate, increasing the likelihood of interactions with shorter wind-driven waves (Gramstad & Trulsen, 2010; Zheng et al., 2016). Higher energy levels in this range indicate the presence of persistent swell contributions, which have been linked to an increased occurrence of rogue waves in bandwidth-limited sea (Støle-Hentschel et al., 2020; Wang et al., 2020). These interactions create an environment where wave groups form and persist for extended periods, a possible precursor to rogue wave development.


The crest-trough correlation parameter (*r*) has emerged as a key indicator for rogue wave occurrences in real-world ocean conditions, as demonstrated in multiple recent studies, including the works of Häfner et al. (2021) and Gemmrich & Cicon (2022). Its strong correlation with rogue wave probability makes it a practical alternative to traditional parameters related to





rogue wave occurrence, enabling more accurate risk assessments for maritime safety. This parameter quantifies the degree of
correlation between the crest heights and successive trough depths in a wave field and has been identified as the dominant
control factor for rogue wave conditions.

The crest-trough correlation $r$ is computed using the auto-correlation function of the sea surface elevation at a lag time of
half the mean spectral wave period (Hafner et al., 2021a). Mathematically, it can be derived from the Wiener-Khinchin
theorem, which links the spectral density of the wave field $S(f)$ to the correlation function:

$$\boldsymbol{r} = \frac{1}{m_0}\sqrt{\rho^2 + \Omega^2}$$

where:

- $m_0$ is the zeroth spectral moment,

- $\rho^2$ and $\Omega^2$ are computed as:


$$\rho = \int_0^\infty S(f)\cos(2\pi f\tau)\,df, \quad \Omega = \int_0^\infty S(f)\sin(2\pi f\tau)\,df$$

where $\tau$ (=$\bar{T}$/2) is the lag time at half the mean period $\bar{T}$, which is estimated as $m_0/m_1$ (where $m_1$ is the first spectral
moment).

This formulation effectively captures the coherence between successive wave crests and troughs, making it highly relevant
for rogue wave prediction. Unlike traditional rogue wave predictors, such as the Benjamin-Feir Index (BFI) and wave
steepness, which are primarily based on modulational instability theories, crest-trough correlation provides a robust
empirical link to rogue wave formation in natural ocean conditions (Gemmrich & Cicon, 2022; Hafner et al., 2021b). High
crest-trough correlation is indicative of strongly correlated wave groups, which enhance the likelihood of large amplitude
waves forming through constructive interference (Gemmrich & Thomson, 2017; Gramstad & Trulsen, 2007; Saulnier et al.,
2011). In contrast to nonlinearity-based metrics (such as BFI), crest-trough correlation seems to remain a statistically
significant rogue wave predictor across all sea states and environments, including deep and shallow water (Hafner et al.,
2021b).

A key aspect of this study is the evaluation of crest-trough correlation as an early warning indicator for rogue wave
development (Cicon et al., 2024; Gemmrich & Cicon, 2022; Hafner et al., 2021b). Since this metric quantifies the degree of
coherence between wave crests and troughs, it serves as a proxy for wave group organization and energy focusing
mechanisms (Hafner et al., 2021a).





This research suggests that rogue waves tend to occur in conditions where crest-trough correlation initially decreases below
0.5 before recovering above 0.6, indicating a restructuring of the wave field from a more chaotic to a more clustered state. To validate this hypothesis, four different rogue wave events were chosen from four strategically positioned CDIP buoy stations: Station 098 (Mokapu Point, HI), Station 154 (Block Island, RI), Station 162 (Clatsop Spit, OR), and Station 166 (Ocean Station Papa). These locations provide diverse oceanic environments for investigating rogue wave formation under varying swell and wind-sea conditions.


The analysis examined how crest-trough correlation evolves alongside spectral bandwidth narrowing and relative energy shifts within the 0.25–1.5 Hz frequency range. Spectral bandwidth narrowing is an essential indicator of energy concentration within fewer dominant frequency modes, enhancing wave coherence and constructive interference (Hafner et al., 2021a). The relative energy distribution in the 0.25–1.5 Hz band highlights the interaction between long-period swell
waves and shorter wind-driven waves, influencing rogue wave growth (Hafner et al., 2021a). By simultaneously analyzing these parameters, we examined how wave field transformations precede extreme wave events. This methodology integrates traditional wave height analysis with advanced spectral diagnostics and crest-trough correlation assessments.

## 3 Results and Discussion

### 3.1 Entire Dataset Analysis for Common Period Time

Our first investigation aimed to identify the seasonal distribution of real-world rogue wave occurrences based on FOWD buoy data (**Fig. 1**). Rogue waves in this dataset have at least 2m height, since FOWD is filtered to only contain wave data with spectral significant wave height above 1m. This analysis highlights clear spatial and seasonal patterns in rogue wave frequency across different regions.

The highest rogue wave occurrences are observed along the U.S. West Coast, particularly offshore California, Oregon, and Washington, where the number of rogue wave detections averages between 80 to 120 events per season in some locations, with peak values exceeding 150 events per season in winter (Baschek & Imai, 2011; Cattrell et al., 2019). The North Atlantic coast, particularly off New England and the Mid-Atlantic, shows moderate rogue wave activity, averaging 40 to 80 events per season at some buoys. In contrast, the Gulf of Mexico and the southeastern U.S. coastal waters exhibit significantly
lower rogue wave activity, with values typically below 20 events per season, and in many locations, rogue waves are rarely detected at all (Jonathan & Ewans, 2011). The Hawaiian region displays moderate occurrences, with values ranging from 30 to 70 events per season, depending on the season. Rogue wave activity near Alaska is less consistent but can reach 50 to 90 events per season during the most active months.







**Fig.1** – Average number of rogue waves per season based on CDIP buoy data from the FOWD dataset from 2015 to 2021. Color gradients represent the average number of rogue waves detected per season at each buoy location.

The seasonal distribution of rogue waves seems to follow a trend influenced by large-scale meteorological and oceanographic processes (Wang & Swail, 2001). Rogue waves are most frequent in the Winter season, with the West Coast experiencing peak values of 100 to 150 events per season in certain locations (Cattrell et al., 2019). The Northeast U.S. and Mid-Atlantic also show increased occurrences, with 50 to 100 events per season at some buoys. This is consistent with the dominance of extratropical storm activity in the North Pacific and North Atlantic, which generates high-energy wave fields conducive to nonlinear wave interactions that form rogue waves. A moderate decline in rogue wave occurrences is observed in the spring. Along the West Coast, values drop to 60–100 events per season, while the East Coast sees a reduction to around 40–70 events per season. This seasonal shift is likely related to the weakening of winter storms and the transition into calmer springtime wave conditions. Occurrences are at their lowest levels across all regions in the summer. The West Coast





records around 40–80 events per season, while the East Coast and Gulf of Mexico rarely exceed 20–40 events per season.
The Hawaiian region still exhibits moderate occurrences, around 30–60 events per season, possibly due to long-period swells generated by distant storms. And in the fall, there is a resurgence in rogue wave occurrences, particularly along the West Coast, where values climb back to 80–120 events per season. The East Coast also sees an increase, with values reaching 40–80 events per season. This seasonal uptick aligns with the intensification of storm activity as the transition to winter begins.

The strong winter peak in rogue wave occurrences, especially along the West Coast and North Atlantic, corresponds to the seasonal intensification of extratropical cyclones, which generate unstable sea conditions, large wave fields and steep wave conditions favorable for rogue wave formation. The summer minimum is expected, as storm activity diminishes, leading to a calmer sea state in general. The fall and spring transitional periods show moderate rogue wave activity, reflecting seasonal shifts in storm intensity and frequency.


Whisker plots were created (**Fig. 2**) to present the monthly distribution of significant wave height (Hs) and maximum individual wave height (Hmax) our three distinct data sources: CDIP-FOWD (buoy observations), ERA5 (reanalysis), and ECMWF CY47R1 (high-resolution hindcast). This provided not only insights into the seasonal variations, but also aimed to show the discrepancies between datasets, and implications for rogue wave identification. The central box represents the
interquartile range (IQR), which contains the middle 50% of the data points (from the 25th to the 75th percentile). The horizontal line inside the box represents the median value of the dataset for each month. The whiskers extend to 1.5 times the IQR or the minimum and maximum non-outlier values. The small hollow black circles represent outliers, which are values that exceed the whisker range and indicate extreme wave events.

The presence of numerous outliers, particularly in Hmax, suggests that extreme wave occurrences are common, especially in winter months. These outliers are particularly relevant because rogue waves are extreme events, and their detection relies on capturing these deviations from the general wave height distribution. The smaller number of extreme outliers (isolated black circles) in ERA5 and ECMWF CY47R1 suggests they do not fully capture rogue waves, supporting previous studies that indicate reanalysis datasets smooth out extremes.


The highest values of Hmax and Hs occur in January, February, and December, with numerous Hmax outliers exceeding 20–25 meters and extreme values surpassing 30 meters, and a few Hs outliers even exceeding 10 meters. The median Hmax during winter months is around 5 to 7 meters in FOWD, compared to 4 to 6 meters in ERA5 and ECMWF CY47R1. The greater spread of the whiskers and the large number of extreme outliers during these months indicate increased extreme wave
events. Hmax and Hs values are significantly lower during summer, with Hmax median values dropping to 3 to 5 meters and Hs values dropping to 1.5 to 2.5 meters across all datasets. The interquartile range is narrower, and fewer extreme outliers are present, suggesting a decrease in storm-driven wave activity. During fall and spring transitions, from September–



November and April–May, there is a gradual increase in Hmax and Hs values and an increase in the values spread, reflecting the seasonal shift toward stormier conditions and higher wave variability.


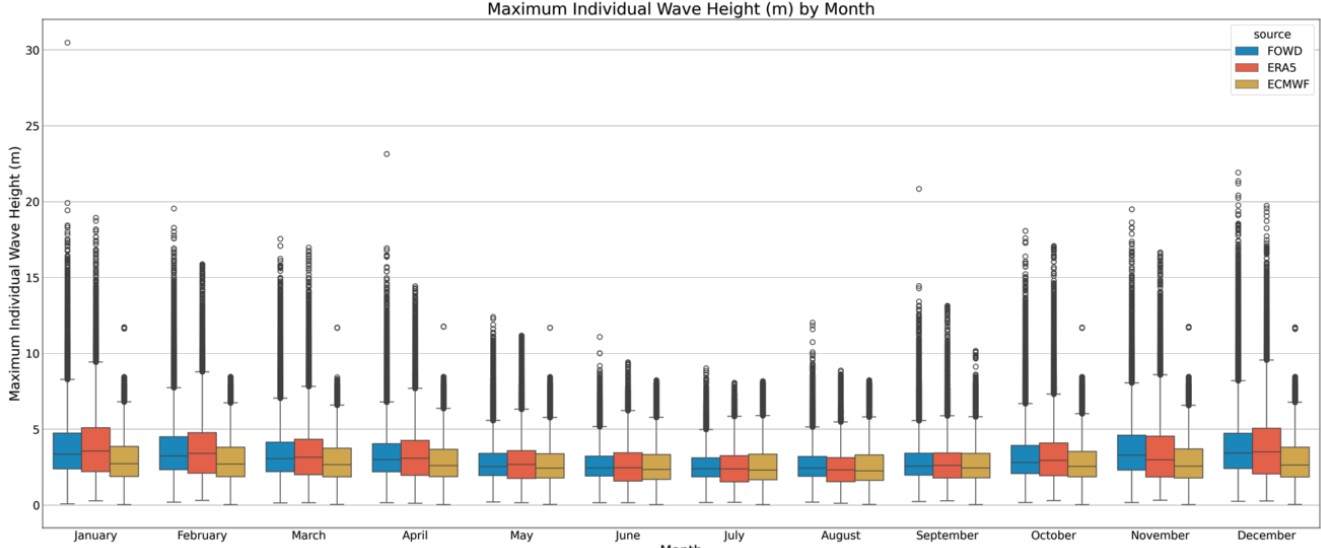

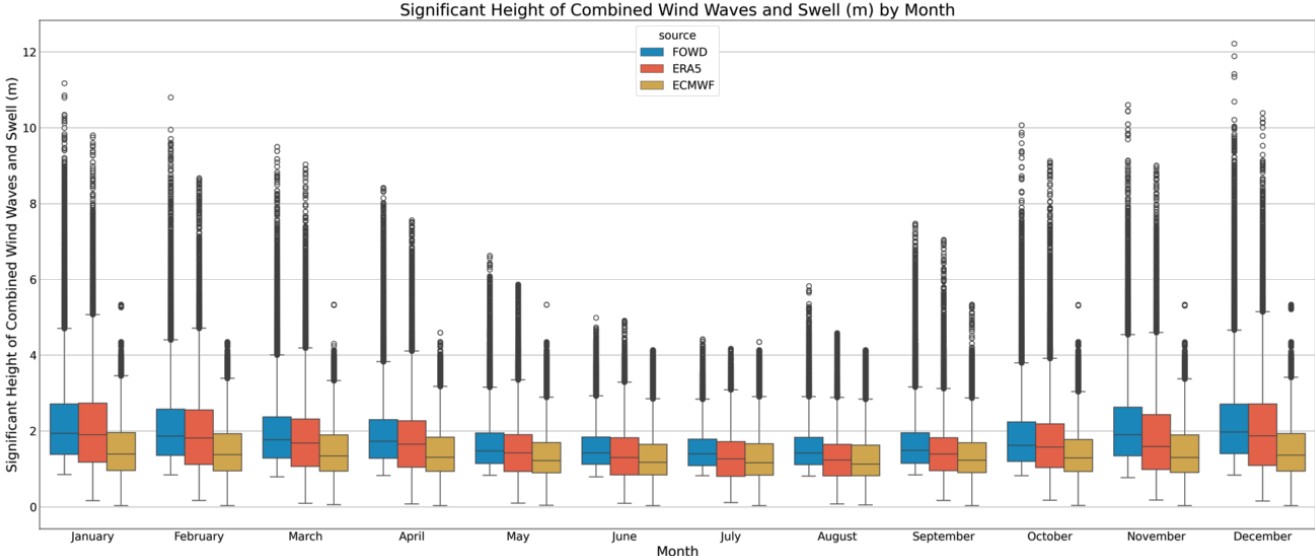

**Fig. 2 -** Whisker plots representing statistics of maximum wave heights (in meters) and significant wave height (in meters) per month for the buoy data (all locations combined from the FOWD dataset), the ERA5 dataset and the ECMWF dataset for

315                                 the same period, from 2015 to 2021.



Comparing the Hmax datasets it is apparent that FOWD captures more extreme events, with the median and IQR of FOWD (blue bars) are consistently higher than those of ERA5 (red) and ECMWF CY47R1 (yellow), particularly in storm seasons
(winter and fall). Although ERA5 captures some large outliers (reaching 15–20 meters), its whiskers and box range are typically narrower, suggesting a smoothing effect in reanalysis data. The ECMWF CY47R1 exhibits the smallest spread, systematically recording lower Hmax values, with median values 0.5–1 meter lower than ERA5 and FOWD, indicating a potential underestimation of extreme wave events. For the Hs data, the FOWD has a broader distribution, capturing more variability in wave height. Median values remain slightly higher than ERA5 and ECMWF, particularly in winter. Both ERA5
and ECMWF CY47R1 exhibit more constrained distributions, with ERA5 showing some high outliers but with a smaller IQR compared to FOWD, suggesting reanalysis data smooths out extreme conditions. ECMWF CY47R1 consistently has the lowest Hs values, with a narrower distribution.

The analysis of Hs and Hmax across FOWD, ERA5, and ECMWF CY47R1 reveals that buoy-based observations (FOWD)
consistently capture larger extreme wave events than the model datasets, particularly in winter months when storms dominate. The presence of numerous outliers in FOWD suggests a more accurate detection of extreme conditions, whereas ERA5 and ECMWF CY47R1 appear to smooth out these events, leading to an underestimation of rogue wave occurrences.

The monthly whisker plots infer that rogue wave occurrences are more frequent when overall wave energy is higher. The
direct correlation between high Hs and extreme Hmax values supports the conclusion that the probability of rogue waves increases during high-energy sea states. Winter and fall periods show the highest wave heights, aligning with the most frequent rogue wave occurrences in the seasonality map. Summer shows the lowest wave heights, with minimal rogue wave occurrences, consistent across both graphs. Spring acts as a transitional phase, with moderate values in both wave height and rogue wave frequency. The fact that the highest median and extreme values in the whisker plots align with peak rogue wave
seasons in the map suggests that rogue waves are not totally random but linked to seasonal storm activity causing higher waves and more instability in the ocean, and regional wave climate conditions. The west coast domination in rogue wave events suggests that high wave height alone is not the only factor for rogue wave formation; local bathymetry, wave-current interactions, and linear and nonlinear effects also likely play a role (Cattrell et al., 2018). This joint analysis emphasizes that certain locations and seasons pose significantly higher risks of rogue wave encounters so mapping rogue waves occurrence
globally per season can make a big difference for maritime planning and safer navigation in general.

Scatter density plots are also a powerful tool for visualizing the relationship between two different datasets (**Fig. 3**) (Cicon et al., 2024). Each point in the scatter plots below represents a pair of values, with FOWD buoy measurements on the x-axis and the corresponding values from ERA5 or ECMWF CY47R1 on the y-axis. The correlation coefficient (r-value) in each
plot quantifies how well the model data aligns with the buoy measurements, with values closer to 1.0 indicating a stronger agreement.





**Fig. 3 -** Density Scatter Plots of ERA5 reanalysis vs. FOWD buoy data and ECMWF CY47R1 hindcast vs. FOWD buoy data for maximum wave height (Hmax), significant wave height (Hs) and rogue wave index (Hmax/Hs) for all buoy locations for the years of 2015 to 2021 (inclusive). The color gradient represents the density of points, where warmer colors (red/yellow) indicate a higher concentration of data points and cooler colors (blue) represent lower densities.





Analyzing the Maximum Individual Wave Height (Hmax) graphs, the ERA5 vs. FOWD graph on the upper left panel shows a correlation coefficient r = 0.91 indicating a strong agreement between ERA5 and FOWD in estimating Hmax. Most data

points cluster around the 1:1 line, particularly for Hmax values below 10 meters, suggesting that ERA5 captures the general trend well in most sea states. However, as Hmax increases beyond 10 meters, ERA5 tends to underestimate extreme values. The density of data points (color gradient) shows the highest concentration of values around Hmax = 2 to 8 meters, reflecting the most common wave conditions. While comparing Hmax between ECMWF vs. FOWD we note the same r=0.91 strong correlation, with a distribution nearly identical to that of ERA5, including the same underestimation more pronounced

beyond 10 meters, where ECMWF values tend to be lower than FOWD buoy measurements. Both models show strong agreement with buoy measurements for moderate sea states, but underestimate extreme Hmax values, which are crucial for rogue wave detection. The smoothing effect in reanalysis and hindcast models likely contributes to this bias, as extreme waves are less frequently resolved.

On the Significant Wave Height (Hs) graphs, both the ERA5 and the ECMWF vs. FOWD graph shows a correlation even higher of r=0.93, suggesting that the models perform slightly better in predicting significant wave height than maximum individual wave height. Small deviations appear for higher Hs values (>6 meters), where the models shows a minor underestimation. Both ERA5 and ECMWF perform well in capturing the significant wave height (Hs) across different sea conditions, but the ECMWF appears to show slightly more spread, suggesting small biases in high-energy wave conditions.

Note that the FOWD code filters and removes all data with spectral significant wave height ($Hm_0$) lower than 1 meter to avoid unsignificant data. On these scatter plots we chose to use the time domain significant wave height (Hs) from FOWD instead of the $Hm_0$, so we can still see a few points with significant wave height below 1 meter.

Looking at the rogue wave detection graphs, calculated from Hmax/Hs, we do not see any correlation between the ERA5 or

ECMWF data with FOWD.  Most ERA5 values cluster between Hmax/Hs = 1.5 to 2.1, showing a bias on this area, while the FOWD data ranges from Hmax/Hs = 0.2 to 3.2. ERA5 appears to filter out the data with Hmax/Hs>2.0, so it heavily underestimates rogue wave occurrences, despite the buoy data indicating higher occurrences. The ECMWF shows a similar pattern to ERA5, however with values clustering between 1.7 to 2.4, a little higher than the ERA5 values. There are a lot more cases of Hmax/Hs > 2.0, which shows that ECMWF detects more rogue waves in comparison to ERA5.


While both ERA5 and ECMWF show strong correlation in estimating Hmax and Hs, their lower estimates of Hmax/Hs suggest that rogue wave events are not considered or less frequently represented in the models. This finding aligns with previous studies that indicate reanalysis and hindcast models struggle to resolve transient extreme wave events (Campos et al., 2018; Hersbach et al., 2020; Lobeto et al., 2024). Both ERA5 and ECMWF average wave conditions over large grid cells

(40 km for ERA5, 18 km for ECMWF), which smooth out extreme wave events detected by localized buoy measurements.



Probability density function (PDF) graphs were additionally created to represent the likelihood of different values occurring within a dataset (**Fig.4**) (Nederkoorn & Seyffert, 2022).

Although all three datasets exhibit a similar shape of PDF distributions for Maximum Individual Wave Height (Hmax), the FOWD dataset exhibits a markedly broader distribution values, with a higher standard deviation and a maximum exceeding 23 meters. This contrasts sharply with the more conservative estimates from ERA5 and ECMWF, which exhibit maximum Hmax values of 16.07 meters and 10.17 meters, respectively. The FOWD data also reveals a longer-tailed distribution, capturing more extreme events that are systematically underestimated by both numerical models. This discrepancy highlights

the tendency of spectral models to smooth localized high-energy wave events, thereby masking short-lived rogue wave occurrences. Similarly, the distribution of significant wave height (Hs) further reinforces this observation. While ERA5 and ECMWF attain reasonable agreement with FOWD in terms of median values, both models exhibit a narrower range and reduced standard deviation, with ECMWF showing a particular deficiency in reproducing storm-driven peak Hs values. The long tail in FOWD's distribution, related to a confirms that buoys capture higher energy wave states that models struggle to

replicate. Both graphs seem to fit well with a Fréchet distribution (heavy-tailed Generalized Extreme Value) with the slow decay of probability towards very high (extreme) values. Such a distribution indicates that events significantly greater than the mean occur more frequently than expected under a Gaussian (normal) or exponential model

  The rogue wave index (Hmax/Hs) PDF is critical since Hmax/Hs > 2 is a defining threshold for rogue waves and this is the

graph that shows some disagreement. At this PDF, the FOWD dataset exhibits a strong peak between 1.6-1.8 which depicts the relationship we usually see in real life, where extreme waves are less common (Nederkoorn & Seyffert, 2022). FOWD also shows a long tail with a broader distribution and with more points exceeding 2, meaning it captures more rogue wave events than ERA5 and ECMWF. Meanwhile, the model distributions cluster around Hmax/Hs ≈ 1.9 to 2.0 on ERA5 and on 1.9 to 2.1 on ECMWF, suggesting some type of model calculation bias which is not always seen in real life (Janssen, 2015).

The outlier's maximum in ECMWF probably occurs due to the significant wave height being smaller than 1 m, which means that Hmax is divided by a value lower than 1, making the ratio value too high, with no physical meaning. The overall narrowness of the model distributions reflects a general challenge to resolve rogue wave-scale amplification processes.

  Wave spectral skewness measures the asymmetry of the wave spectrum, characterizing how energy is distributed across

wave frequencies (Stansell, 2004). High skewness values are often linked to wave focusing mechanisms, which amplify extreme events. Previous research (Mori & Janssen, 2006) confirms that skewness is a good predictor of rogue wave formation, particularly in shallow water. The FOWD dataset shows a broader distribution, capturing more variability in wave spectra, while both ERA5 and ECMWF are highly constrained around 0, meaning they do not represent well the asymmetry in wave spectra, a key mechanism for rogue wave growth.





**Fig. 4** – Probability Density Functions of different wave parameters comparing data from buoys (FOWD dataset), depicted in blue, the ERA5 reanalysis depicted in orange and the ECMWF CY47R1 hindcast depicted in green for the same period, from 2015 to 2021. The zoom of the positive tail is depicted at the upper right. Basic Statistics are shown on the tables.


The same is seen with kurtosis, which measures the "peakedness" of the wave energy spectrum, indicating how concentrated wave energy is within specific frequencies (Goda, 1970). Higher kurtosis values have been linked to increased rogue wave probabilities, as they indicate wave focusing mechanisms (Mori & Janssen, 2006; Mori et al., 2011). Theoretical work (Fedele et al., 2016) suggests that spectral peakedness (high kurtosis) is a necessary condition for rogue waves to form. The FOWD dataset exhibits a broader range, suggesting that the buoy data can better capture extreme spectral peaking and that

the models are less effective at capturing it.

FOWD data demonstrates both more positive skewness and elevated kurtosis values, indicating frequent departures from Gaussian behavior in the wave field. These are signatures of asymmetric, steep, and strongly peaked wave groups, conditions known to precede rogue wave events. In contrast, both ERA5 and ECMWF exhibit distributions that are centered near zero

with low variability, highlighting their lack of sensitivity to wave shape irregularities and spectral sharpness.

The Benjamin-Feir Index (BFI) is a well-known predictor of modulational instability. Theoretical studies (Janssen, 2003; Onorato et al., 2005) indicated that modulational instability is a dominant mechanism for rogue waves in deep water. Higher BFI values (>1) indicate a greater likelihood of wave group instability, which can possibly lead to rogue wave generation.

The FOWD dataset again shows a broader distribution (the maximum outliers from the models should be ignored), capturing occasionally high BFI values, especially with its long tail. Both ERA5 and ECMWF distributions are tightly constrained around 0, meaning they do not represent modulational instability well. This aligns with previous research showing that models relying on spectral wave parameterizations often do not capture the nonlinear physics necessary for rogue wave formation (Janssen, 2015).


Collectively, these findings confirm that, as expected, FOWD buoy measurements provide a more robust representation of real-world wave variability, including extreme events and rogue wave precursors. The systematic underestimation of Hmax and Hmax/Hs, and the absence of spectral asymmetry and peakedness in ERA5 and ECMWF, reflect fundamental limitations in model physics and resolution. These limitations have direct implications for operational wave forecasting and

rogue wave warning systems.

**3.2 Specific Rogue Wave Events Analysis**

Recent advances in ocean wave dynamics, particularly the work of Dion Häfner et al. (2021), emphasize the importance of additional spectral parameters in improving rogue wave prediction and classification (Hafner et al., 2021b). These include spectral bandwidth narrowness, the relative energy contained within the 0.25–1.5 Hz range, and crest-trough correlation. By

incorporating these parameters into the analysis, it is possible to gain a more refined understanding of wave evolution and the mechanisms leading to rogue wave amplification.



In this study, we looked at four rogue wave occurrences in four different CDIP buoy stations to more deeply investigate the sea state evolution before, during and after a rogue wave event. The buoys' FOWD data parameters of interest were mainly
compared to the ECMWF CY47R1 wave hindcast data from 2015 to 2021, which also included the global values for crest-trough correlation, relative energy on the 0.25 to 1.5 Hz frequency and the narrowness spectral bandwidth based on Hafner's formulas. For the more usual parameters maximum wave height, significant wave height and the rogue wave index, which is just one divided by the other (Hmax/Hs), we also compared with the ERA5 reanalysis data and the non-filtered CDIP data, directly from calculations from the heave (surface elevation) raw data from the CDIP archives. The selected buoys—
Mokapu Point, HI (098), Block Island, RI (154), Clatsop Spit, OR (162), and Ocean Station Papa (166)—are positioned in distinct oceanic regions, providing datasets for understanding the conditions that contribute to rogue wave events in different environments (**Fig. 5**).

The North Pacific, represented by Ocean Station Papa (166), is characterized by persistent long-period swells and high-
energy wave climates, a good location for studying nonlinear wave interactions. Block Island (154) can be influenced by extratropical storm activity, providing an opportunity to examine how synoptic-scale weather systems impact rogue wave generation. Clatsop Spit (162), positioned along the Pacific Northwest, is subject to both local wind-driven waves and open ocean swells, offering a complex wave climate for spectral analysis. The Mokapu Point buoy (098) in Hawaii captures wave data in a region where energy is frequently influenced by trans-Pacific swell propagation and episodic storm activity.


We first note that in the four locations graphs the calculations based on the raw, non-filtered, CDIP surface elevation data (in red) usually does not match 100% with the filtered CDIP data calculation from the FOWD dataset (in green). When using the raw, non-filtered, data, we sometimes see rogue waves peaks (when data goes above 2 on the Hmax/Hs graphs) that were not present on the filtered data. For future studies, it is important to check if perhaps the filter CDIP and others are using are
not unintentionally removing real rogue waves events, which could be perceived as non-real outliers, from the data. The usual modelled data (ERA5 or ECMWF) smoothing can also be clearly noticed in all the graphs from **Fig.5**.

Observational data from the selected rogue wave events at stations 098, 154, 162, and 166 consistently demonstrated that a decrease in crest-trough correlation was accompanied by a simultaneous narrowing of spectral bandwidth and an increase in
relative energy within 0.25–1.5 Hz. This suggests that these spectral changes occur in tandem with the restructuring of the wave field, marking a transition phase where rogue waves are more likely to emerge. The spectral focusing effect, driven by the narrowing of bandwidth and redistribution of energy toward lower frequencies, leads to an increase in wave amplitude variability and supports the formation of extreme, isolated waves.




**Fig. 5** – Specific rogue wave event analysis at four different buoy stations (098, 154, 162, and 166) during a period of 72 hours. ERA5 data is depicted in blue, ECMWF CY47R1 data depicted in orange, FOWD data depicted in in green and raw CDIP data depicted in red.





## 4 Discussion

The crest-trough correlation parameter is strongly linked to spectral bandwidth, with narrower bandwidths leading to more correlated wave structures(Cicon et al., 2024). This relationship is crucial because wave groups in bandwidth-limited conditions favor rogue wave development. Studies using buoy observations from the FOWD dataset and wave modeling with WAVEWATCH III (WW3) show that high crest-trough correlation values (>0.6) are associated with rogue wave probabilities that are an order of magnitude higher than in uncorrelated wave fields (Cicon et al., 2023). Häfner et al. (2021) showed that rogue wave probability of occurrence varies by a factor of 10 based on crest-trough correlation alone, far exceeding the indicator capability of parameters like kurtosis, skewness, or steepness. And unlike kurtosis, which is only useful within single wave groups, crest-trough correlation remains a reliable predictor across extended time periods and different oceanic regions. Since crest-trough correlation can be calculated from wave spectra moments, it can be directly implemented into operational wave forecast models like those run by ECMWF and NOAA.

The link between spectral bandwidth narrowness, relative energy and the crest-trough correlation parameter can be understood through their combined impact on wave evolution. As spectral bandwidth narrows, wave energy is confined to fewer dominant modes, reducing spectral dispersion and increasing the likelihood of constructive interference. This effect is further amplified when energy in the 0.25–1.5 Hz range increases, signifying a shift toward wave fields where swell components reinforce the background wave spectrum rather than dissipating across a broader range of frequencies. This process results in enhanced wave group formation, where wave trains become more phase-aligned over time, increasing the crest-trough correlation values and consequently the probability of extreme wave occurrences.

Häfner's FOWD introduction article (Hafner et al., 2021a) contains a probability density function (PDF) graph for crest-trough correlation parameter *r* for all FOWD waves. It shows a distinction in correlation values between all waves, waves with Hmax/Hs > 2 (moderate rogue waves), and waves with Hmax/Hs > 2.4 (extreme rogue waves). For all waves, the crest-trough correlation (*r*) values are broadly, almost normally, distributed across a range spanning approximately 0.2 to 0.9, with a peak around 0.6 to 0.7. For waves that have a height that is more than twice the significant wave heigh, the crest-trough correlation distribution shifts a bit toward higher values, with most cases occurring above 0.5, and the peak moving toward 0.7 to 0.8. And for extreme rogue waves (rogue wave index> 2.4), the crest-trough correlation range is even more constrained, with most values falling between 0.6 and 0.9, and a pronounced peak around 0.75 to 0.85. This strong shift toward higher correlation values confirms that rogue waves tend to form in conditions where wave crests and troughs are highly correlated, reinforcing the role of linear superposition in rogue wave generation. On the other hand, this PDF clearly shows that it is not possible to consider a 0.6 or 0.7 crest-trough correlation parameter threshold alone to identify rogue waves since in general seas the values go up to 0.9 with most values around 0.6.




While it is well established that rogue waves are better sustained in sea states where the crest-trough correlation remains
above 0.6 due to wave groupness and focusing, this alone does not serve as an effective predictor. Instead, it is the preceding
conditions—a temporary drop in crest-trough correlation followed by a rapid increase—that provide a more reliable early
warning indicator. This finding suggests that rogue waves tend to emerge in dynamically evolving sea states rather than in
purely steady conditions. For a rogue wave warning system, the focus should not be solely on identifying when crest-trough
correlation exceeds 0.6, but rather on recognizing the transition phase that precedes it. By monitoring the evolution of crest-
trough correlation in combination with spectral bandwidth changes and relative energy shifts, it may be possible to establish
a probabilistic framework for forecasting rogue wave risk.

Across all four locations on **Fig.5**, the evolution of this parameter follows a distinct sequence where an initial decrease in
crest-trough correlation below 0.5 is observed, followed by a rapid increase exceeding 0.6 just prior to or during the rogue
wave event. This pattern strongly suggests that rogue wave conditions are preceded by a transitional phase in the wave field,
where wave coherence temporarily weakens before re-emerging in a more clustered and structured state, favoring the onset
of extreme wave amplification.

At Station 154 (Block Island, RI), a pronounced rogue wave event occurs on November 16, 2018, with a sudden spike in
Hmax exceeding 20 meters. Prior to this event, crest-trough correlation drops below 0.5, followed by a rapid recovery above
0.6. Concurrently, spectral bandwidth narrowness increases, indicating that wave energy is being redistributed into fewer
dominant frequencies, a process that aligns with constructive interference mechanisms. The rogue wave event is marked by
an increased crest-trough correlation, reinforcing the idea that high correlation values (>0.6) sustain rogue waves, but the
transition from a lower correlation state is what signals their imminent formation.

A similar pattern is evident at Station 162 (Clatsop Spit, OR) during the rogue wave occurrence on April 8, 2018. The crest-
trough correlation parameter initially drops sharply, reaching a minimum near 0.45, before rising to values exceeding 0.65
just as the rogue wave event occurs. This sequence is accompanied by an increase in relative energy within the 0.25–1.5 Hz
range, suggesting that swell-wave interactions are playing a significant role in rogue wave formation at this location. The
relationship between these spectral characteristics and the evolving crest-trough correlation further supports the idea that
rogue waves form in dynamically changing sea states rather than in static conditions with consistently high correlation.

At Station 166 (Ocean Station Papa), the rogue wave event on December 30, 2015, follows the same pattern, with crest-
trough correlation first dropping below 0.5, signaling a reduction in wave coherence, and then recovering above 0.6
immediately before the rogue wave appears. This behavior is accompanied by increasing spectral bandwidth narrowness and
rising relative energy levels, reinforcing the finding that the restructuring of wave groups precedes rogue wave formation.



These observations confirm that while rogue waves are better sustained in seas where crest-trough correlation remains above 0.6, which means this is a robust rogue wave identifying condition, however this parameter alone is not a sufficient predictor. Instead, the transition phase, where crest-trough correlation temporarily decreases before increasing again, appears to be a more reliable precursor to rogue wave formation.

In order to statistically verify the accuracy of this inverted peak, or drop on the crest-trough correlation, hypothesis in relation to rogue wave identification, we performed an analysis using the data from the FOWD quality-controlled dataset. Data available for stations 154, 162, 166 and 098 between 2015 and 2021 were gathered, and the total number of rogue waves was found simply checking the relative height parameter being larger than 2. This parameter is the calculated Hmax divided by the spectral significant wave height every 30 minutes. There were 81 rogue waves found. Note that the FOWD dataset is filtered to contain only waves with significant heights above 1m, which translates to rogue waves with a 2m height minimum. Then we downloaded the date related crest-trough correlation parameter from 1.5 days before and 1.5 days after the located rogue wave event. During this time, we looked for the inverted peak that had a minimum 0.1 difference in the crest-trough correlation values and that reached a correlation of 0.5 or below. We found 61 instances that agreed with our hypothesis, which means that 75.3% of the rogue waves followed our criteria. This suggests that a real-time rogue wave warning system should focus on detecting this sharp drop and recovery in crest-trough correlation, rather than merely identifying when values exceed a threshold.

Cicon et al. (2024) proposed an empirical formula which initially expressed rogue wave occurrence probability as a function of certain parameters, including wave steepness, relative depth, directional spread, and spectral bandwidth, alongside with the crest-trough correlation (*r*) parameter. However, after evaluating the performance of this formula using buoy observations and ECMWF hindcast data, they found that the inclusion of these additional variables offered only limited improvement in predictive skill, since the coarse resolution of global models usually fail to accurately capture nonlinear effects, particularly in shallow water environments, where these parameters are expected to play a more significant role. So, Cicon et al. (2024) computed a simplified version of the empirical formula in which steepness, relative depth, directional spread, and spectral bandwidth are replaced with their mean values, reducing the equation to a function of only *r* and two empirical constants (Cicon et al., 2024). The resulting expression:

$$\mathbf{P} \ (H>2Hs) = f(\textit{r}) = a_0 \ e^{(3.8r + a_1)}$$

where $a_0 = 1.37$ and $a_1 = -12.12$ which is a purely correlation-based predictor of rogue wave probability. This equation exhibited an even stronger predictive power (72%) than the initial equation when compared to buoy data (Cicon et al., 2024), however this number is still lower than the predictive power calculated from our inverted peak hypothesis.

While Cicon et al. equation showed an improved fit to in-situ data, we propose that its predictive power can be improved if, instead of considering only the absolute value of $r$ at a given moment in time, the models also account for its temporal evolution. Our goal is to work on the implementation of this model for our future work. Our results agree that crest-trough correlation $r$ is the dominant control factor in rogue wave occurrence probability, however its temporal evolution and spectral context should be considered for an operational early warning system.

## 5 Conclusions

This study provides an initial assessment of wave models and rogue wave occurrences by comparing in situ observations from FOWD (Filtered Ocean Wave Data) buoys with model-based estimates from ERA5 reanalysis and the ECMWF CY47R1 high-resolution hindcast. By integrating multiple analyses, including seasonal rogue wave distributions, statistical comparisons of maximum wave heights (Hmax) and significant wave heights (Hs), and density scatter plots with additional spectral parameters (skewness, kurtosis, and BFI), we identified critical differences in how rogue waves are represented across these datasets. Furthermore, this research aims to validate the effectiveness of crest-trough correlation as a leading indicator of rogue wave risk while evaluating the role of spectral bandwidth and energy distribution in wave amplification processes.

A map analysis of rogue wave occurrences from the FOWD dataset revealed a strong seasonal dependence, with peak rogue wave activity occurring in winter and fall and the highest concentrations of rogue waves found along the West Coast and then the North Atlantic. This corresponds to the seasonal intensification of extratropical cyclones in these areas. The Gulf of Mexico and the southeastern U.S. coastline exhibit significantly lower rogue wave occurrences, likely due to a calmer wave climate.


The monthly whisker plots of Hmax and Hs strongly supports these seasonal findings. Maximum wave heights and significant wave heights peak in winter, with median Hs values of combined wind waves and swells reaching 2.5–3.5 meters and extreme Hmax values exceeding 25–30 meters. The presence of numerous outliers in winter months suggests a greater probability of rogue wave formation during storm-driven high-energy sea states. Summer, on the other hand, shows the lowest wave activity, with median Hs values of 1.5–2.5 meters and Hmax values between 3–5 meters, corresponding to a minimum in rogue wave occurrences. Further studies pursuing the analysis of the regionalization of this data are recommended.

The model data from ERA5 reanalysis and ECMWF CY47R1 hindcast systematically underestimated extreme wave events, particularly for higher maximum individual wave heights (Hmax). While both models exhibit strong correlations with FOWD measurements ($r \approx 0.91$–0.93 for Hmax and Hs), their distributions show a consistent bias toward lower values. The



density scatter plots confirm that ERA5 and ECMWF align well with FOWD for moderate sea states (Hmax < 10 m) but significantly underestimate extreme values (>15 m). This suggests that model smoothing and spatial averaging limit their ability to resolve transient, high-energy wave events, which are critical for rogue wave detection. The whisker plots further
emphasize this underestimation, as FOWD consistently records higher median and extreme Hmax values than both models, especially in storm-dominated months. ECMWF CY47R1 exhibits a narrower range of variability compared to ERA5, possibly due to its reliance on pure model simulations without data assimilation from buoys.

The buoys FOWD data frequently records a lot more rogue wave occurrences if compared to ERA5 and ECMWF model
data. The scatter plots of Hmax/Hs reveal that both models cluster tightly around values near 2.0, whereas FOWD displays a broader distribution with a higher incidence of Hmax/Hs > 2.5, demonstrating that rogue waves are significantly underrepresented in reanalysis and hindcast datasets. This underestimation may be linked to the smoothing of extreme values and limited representation of nonlinear wave interactions in model parameterizations (Campos et al., 2018; Lobeto et al., 2024).


A few spectral parameters recognized to influence rogue wave formation were also analyzed on probability density functions: wave spectral skewness, spectral kurtosis, and the Benjamin-Feir Index (BFI). The buoys FOWD data exhibited a broader skewness, kurtosis and BFI distribution, capturing more variability in wave shapes compared to ERA5 and ECMWF, which cluster tightly around zero, indicating a lack of asymmetry, peakedness and instability in modeled wave
spectra. These findings align with previous studies that suggest spectral wave models often rely on linear approximations, limiting their ability to resolve extreme wave dynamics (Janssen, 2015).

The underestimation of Hmax, rogue wave occurrences, and nonlinear wave properties in ERA5 and ECMWF suggests that reliance on reanalysis and hindcast data alone may lead to underpredictions of extreme wave hazards. This is particularly
concerning for the shipping industry, offshore energy platforms, and coastal infrastructure, where rogue waves pose serious risks (Bitner-Gregersen, 2015). The strong seasonal patterns identified in the FOWD dataset emphasize the need for seasonally adjusted forecasting models, particularly for high-risk regions such as the West Coast of the U.S. and the North Atlantic.

Lastly, specific rogue wave occurrences across four distinct locations were analyzed - Station 098 (Mokapu Point, HI), Station 154 (Block Island, RI), Station 162 (Clatsop Spit, OR), and Station 166 (Ocean Station Papa) from CDIP buoys. Across all stations, rogue waves were consistently preceded by a sharp drop in crest-trough correlation below 0.5, followed by a rapid recovery exceeding 0.6, indicating a transition from a less organized wave field to a more clustered and structured state. This pattern was accompanied by an increase in spectral bandwidth narrowness and relative energy in the 0.25–1.5 Hz

range, suggesting that energy redistribution and constructive interference mechanisms play a significant role in rogue wave formation (Gemmrich & Cicon, 2022; Gemmrich & Thomson, 2017; Hafner et al., 2021b).

A statistical evaluation of this novel crest-trough correlation inverted peak hypothesis to identify rogue waves using FOWD buoy data showed a 75.3% agreement. These findings reinforce previous studies that have highlighted crest-trough

correlation as a dominant indicator of rogue wave probability, yet they also extend this understanding by demonstrating that the absolute value of the crest-trough correlation parameter should not be used as warning sign, but rather its dynamic evolution over time. While rogue waves are more likely to be sustained in environments where crest-trough correlation remains above 0.6, it is the transition from low to high correlation that signals their imminent formation. This provides a new perspective on how rogue wave forecasting could be approached, moving beyond static threshold-based indicators to a more

dynamic assessment of wave evolution.

This study also highlights the need for more studies and forecasts to incorporate crest-trough correlation, narrowness spectral bandwidth and relative energy distributions as key parameters in rogue wave forecasting models. By integrating these parameters into the analysis, it is possible to refine forecasting methodologies and improve the understanding of rogue wave

behavior across multiple oceanic environments. Additionally, by mapping these variables on a global scale using high-resolution wave models, it may be possible to identify regions where wave conditions are primed for rogue wave formation. These results from this study contribute to enhanced maritime safety, optimized offshore operational planning, and improved predictive models for extreme wave events.

Future studies should explore the operational integration of the crest-trough correlation parameter (r) to high-resolution wave models such as WAVEWATCH III and ECMWF operational forecasts. This would help identify rogue wave-prone areas based on spectral signatures and crest-trough correlation trends. Additionally, machine learning techniques can enhance predictive capabilities by assimilating large-scale model outputs with observed rogue wave occurrences, enabling the development of a probabilistic rogue wave forecasting system.


**Acknowledgements**

Extensive thanks are given to Dr. Jean-Raymond Bidlot from the European Centre for Medium-Range Weather Forecasts (ECMWF) for his help on providing the higher resolution hindcast data.

**Authors Contributions**

Author Contributions: L.A. was responsible for conceptualization, methodology, software, data analysis and original draft preparation. G.M. assisted with software development and figures. S.M. was responsible for reviewing and editing the



manuscript. M.L. was responsible for resources, project administration, funding acquisition and reviewing and editing the manuscript. All authors have read and agreed to the published version of the manuscript.


**Funding**

This study was funded in part by the Greater Tampa Bay Marine Advisory Council-PORTS, Inc. (a consortium of regional maritime interests), the South-East Coastal Ocean Observing Regional Association (SECOORA), the Gulf of Mexico Coastal Ocean Observing System (GCOOS), the Regional Associations under the US Integrated Ocean Observing System 705 (IOOS) and the University of South Florida College of Marine Science endowed fellowship funds.

**Data availability**

The ERA5 reanalysis dataset can be found at:

https://cds.climate.copernicus.eu/datasets/reanalysis-era5-single-levels?tab=download

The ECMWF CY47R1 hindcast dataset can be found at:

https://doi.org/10.21957/y03s-tz09; https://doi.org/10.21957/strn-cs36; https://doi.org/10.21957/dgkx-1485;

https://doi.org/10.21957/t3vj-b111

The CDIP Buoys FOWD dataset can be found at:

https://sid.erda.dk/public/archives/969a4d819822c8f0325cb22a18f64eb8/published-archive.html


**Competing Interests**

The authors declare no competing interests.

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
