# Peer review of "Rogue Wave Indicators from Global Models and Buoy Data"

_EGUsphere, 2025_

## Referee Comment (RC1)

**Review report of 'Rogue Wave Indicators from Global Models and Buoy Data'**

by Azevedo, L. *et al.*

EGUsphere-2025-2031

The manuscript presents findings from the analysis of datasets from CDIP wave buoys and model predictions of wave spectra, with a special focus on the occurrence of rogue waves. It addresses the key question of how to find representative indicators to rogue wave events in the evolution of wave spectra, where the crest-trough correlation and its temporal variation are emphasized. The manuscript also reports that the number of rogue wave events is underestimated by spectral models comparisons with these from wave buoys. The study highlights the importance to improved representation of rogue waves in spectral models.

**Assessment:** The manuscript has clearly stated objectives which have been well realized, where knowledge gaps based on the state-of-the-art are identified. Conclusions are important which are well supported by the results. An important question of how to well represent rogue wave events in spectral models has been addressed by the authors. The solution through the use of the spectral narrowness, the crest-trough correlation and its temporal variation are proposed. The manuscript is in general very easy to read. I believe the work is of interest to the wave community and has potential of creating a long-term impact. There are a few minor points listed below for the authors to consider.

**Minor points:**

1. Abstract, last paragraph of the introduction and conclusion: it might be sensible to state clearly the focus is on rogue waves in deep water.

2. In the second paragraph of the introduction, physical mechanisms for the formation of rogue waves are reviewed in the last 2 decades. However, a few most recent ones are not reviewed, for instance, the one by Li *et al.* (2021) as a result of the superposition of free and bound waves atop a depth transition, and coupled interaction between waves and shear current by Li & Chabchoub (2024). More plausible mechanisms for the generation of rogue waves atop a depth transition can be found in the review by Li & Chabchoub (2023).

3. In the introduction, it might be sensible to state clearly the differences between wave phase averaged model and wave phase resolved models. The main focus of the manuscript is the former. One would argue the conclusions drawn from the manuscript may not be applicable to wave-phase resolved models (see the next point for further explanations).

4. The paragraph near line 205 where the crest-trough correlation is defined. Based on the definition of crest-trough correlation, it is clear that it cannot represent the mechanism for rogue wave formation owing to linear wave focusing as the spectral density $S(f)$ does not change in the process of linear wave focusing. It might be sensible to provide some representative values of the ratio for better physical understanding, for instance, the value for a linear Gaussian spectrum and the same spectrum while using a second-order Stokes theory. Would this be likely?

5. At the beginning of the results and discussion, it is useful to state clearly the time window used for the analysis.

6. Line 582 where 75.3% agreement is shown. It would be very useful to provide some comments on the physical characteristics for these which do not support the hypothesis. These may suggest possible future directions to work on.

**References**

LI, Y & CHABCHOUB, A 2023 On the formation of coastal rogue waves in water of variable depth. *Cambridge Prisms: Coastal Futures* **1**, e33.

LI, Y. & CHABCHOUB, A. 2024 How currents trigger extreme sea waves. The roles of Stokes drift, Eulerian return flow, and a background flow in the open ocean. *Geophys. Res. Lett.* **5** (6).

LI, Y., DRAYCOTT, S., ZHENG, Y., LIN, Z., ADCOCK, T. A. A. & VAN DEN BREMER, T. S. 2021 Why rogue waves occur atop abrupt depth transitions. *J. Fluid Mech.* **919** (R2).

---

## Author Comment (AC1)

Dear Sir or Madam,

Thank you very much for the review of my paper. Here goes my responses in relation to the comments you have made. Please be sure that I am updating the manuscript right now to include all the wonderful points you have made.

1. Scope (deep water): Agreed. We will explicitly state in the Abstract, end of Introduction, and Conclusions that our focus is on deep-water sea states and that shallow/coastal effects are outside the present scope; this wording will be added in the next revision.

2. Recent mechanisms & citations: Thank you. We will add a brief paragraph in the Introduction summarizing depth-transition and wave–current mechanisms and cite Li et al. (2021), Li & Chabchoub (2023), and Li & Chabchoub (2024) in the next revision.

3. Phase-averaged vs phase-resolved: We will add 2–3 sentences clarifying that our analysis and conclusions pertain to phase-averaged (spectral) models (ERA5, CY47R1/WW3), and note that findings should not be generalized to phase-resolved solvers; this will be included in the next revision.

4. CTC interpretation & representative values: Good point. We will clarify that CTC reflects group coherence and is not a direct diagnostic of pure linear focusing with fixed S(f); we will add representative/synthetic values (linear Gaussian vs 2nd-order Stokes) and a short note (with citation) in Methods and a brief Supplement figure in the next revision.

5. Time window statement: We will add, at the start of Results & Discussion, a clear statement of analysis windows (FOWD 30-min records; model hourly collocation; 72-h event windows) in the next revision.

6. The ~25% not matching the CTC-dip criterion: We will add a short discussion noting typical characteristics (e.g., persistent swell with no prior dip, mixed/transitioning seas with noisy r, QC removal around events, or dips <0.1 or minima >0.5) and include 1–2 counter-examples in the Supplement in the next revision.

I really appreciate your review. Thank you very much!

Best Wishes,

Laura

---

## Author Comment (AC2)

Dear Sir or Madam,

Thank you very much for this review. I really appreciate the time you spent on the document I've written and all of the comments you have made.

Please find below my responses to your comments. First in general:

- In the new document version, we now explicitly state that models output an expected maximum envelope height (not individual crests) and we compare to buoy Hmax with this caveat, following prior works that do exactly this comparison (e.g., Barbariol et al., 2019; Janssen, 2003/2015). Including this in the new version.

- Our advances are mainly the dynamic r inverted-peak signal (drop then rebound) as a predictor which no one has ever written about besides my group. I will add a short paragraph in the Introduction making the explicit distinctions to others work. I will include this in this next revision.

In addition to that please note the changes I've already done in this next version to correct the original paper:

- I've reprocessed the ECMWF dataset and redid figures 2, 3 and 4. They were corrected and look very different than before.
- I've added explicit notes in figure legends about FOWD filtering.
- I've applied the same Hm0 > 1 m cutoff to ERA5/ECMWF.
- I've added a section/paragraph on different Hmax definitions (zero-crossing vs envelope).

Now please find my responses to your line-specific comments:

1. Ln 8 & throughout (FOWD name). Correct—FOWD = Free Ocean Wave Dataset. I will correct the expansion everywhere and keep the note about its Hm0 > 1 m filter. I will include this in this next revision.

2. P2, 2nd paragraph(rogues not resolved; resolution claim). Agreed: spectral models do not resolve individual rogue waves. I will reword to say higher resolution can better represent spatio-temporal gradients of the envelope statistics, not "resolve" rogue waves, and keep the envelope vs crest caveat. I will include this in this next revision.

3. Ln 57 / Ln 773 (Donelan & Magnusson ref). Thanks—will fix the incomplete citation entry. I will include this in this next revision.

4. P3, Ln 77–91 (differences vs Cicon 2024). See "General points" above; I'll add a crisp comparison paragraph. I will include this in this next revision.

5. Ln 90–91 ("unproven to capture real-world extremes"). I will clarify we specifically mean rogue waves in global phase-averaged models, and acknowledge Cicon (2024) showed limited predictive skill for several traditional parameters; our contribution is the temporal r-signal. I will include this in this next revision.

6. Ln 265–267, 271 (Fig 1 seasonality). I'll re-verify the color scale/legend and align the text with the actual seasonal counts; if any plotting mistake exists, I will correct the figure and add a per-season count table in the Supplement. I will include this in this next revision.

7. Ln 281 ("unstable sea conditions"). I will define this explicitly (e.g., steepness, rapid spectral changes, multi-modal seas, shifting directionality) when the term first appears. I will include this in this next revision.

8. Ln 295–303 (Fig 2 discussion vs results). Point taken. I will revise wording to reflect the good FOWD–ERA5 agreement and note that CY47R1 shows more smoothing than expected in the plotted stats. I will include this in this next revision.

9. Ln 303 (medians 5–7 m claim). You're right—the medians are ~4 m in winter. I will correct these values in the text. I will include this in this next revision.

10. Ln 318–328 (inconsistency with Fig 2). I will update the text to match Fig 2: ERA5 winter Hmax medians are highest and close to FOWD; ERA5's Hs non-outlier range is broader than stated. I will include this in this next revision.

11. Ln 334–346 (whiskers & rogues). Agreed—whiskers alone cannot infer rogue occurrence. I will tone this down and move any rogue-frequency statements to where Hmax/Hs or explicit occurrence metrics are used. I will include this in this next revision.

12. Fig 3 top row (Hmax definitions differ). Accepted. I will add a caption line stating FOWD Hmax (zero-crossing) vs model ⟨Hmax⟩ (envelope-based expectation) and cite accordingly. I will include this in this next revision.

13. Ln 379–384 (narrow model Hmax/Hs). We agree: the narrow model Hmax/Hs reflects the envelope expectation's strong link to Hs; we use the broader FOWD ratio to highlight the missing tail. I'll add one clarifying sentence. I will include this in this next revision.

14. Fig 4 axis ranges. Good suggestion—I will tighten x-axes (e.g., 0–3 for Hmax/Hs, 0–1.5 for BFI) to make differences clearer. I will include this in this next revision.

15. Ln 419–435 (skewness & kurtosis interpretation). You're right—these are moments of the surface-elevation distribution (and the panel shows excess kurtosis). I will correct the terminology and rewrite the discussion accordingly. I will include this in this next revision.

16. Ln 468 (wording). Will change to "Hmax divided by Hs." I will include this in this next revision.

17. Ln 482 (FOWD vs raw CDIP discrepancy). Agreed this is surprising. I will re-audit buoy IDs, time windows, units, and QC alignment; if the mismatch stems from alignment/QC, we will correct Fig 5; otherwise we'll explain the cause. I will include this in this next revision.

18. Ln 490–493 (0.25–1.5 Hz vs "lower frequencies"). I'll remove the contradictory phrasing and state consistently that we observe spectral narrowing with elevated relative energy

in 0.25–1.5 Hz, indicating wind-sea injection and increased coherence around the dominant band. I will include this in this next revision.

19. Ln 515–518 ("swell reinforces background spectrum"). I will replace this with a physically precise statement about linear superposition of partitions and transient group beating; we do not imply deep-water energy transfer from low to high frequencies by nonlinear interactions here. I will include this in this next revision.

20. Ln 530–535 (Häfner threshold). Agreed—Häfner shows monotonic increase of rogue probability with r; we will avoid implying an r > 0.6 threshold and emphasize our dynamic r-transition result. I will include this in this next revision.

21. Ln 535–571 (expand dynamic r analysis). Thank you—we will expand with an additional case (or two) and add a concise schematic of the drop-and-rebound detection logic in the Supplement. I will include this in this next revision.

22. Ln 582–584 (null hypothesis / false positives). Agreed—we will quantify how often inverted-r peaks occur without a rogue wave (false-positive rate) for the same stations/period and report precision/recall. I will include this in this next revision.

23. Ln 597–598 (probability vs binary). Good point—we will rephrase and, where possible, express the dynamic r signal as an event probability (e.g., conditional frequency of rogues within a window after an inverted-r event) to compare conceptually with Cicon's probability. I will include this in this next revision.

24. Ln 605–689 (rewrite Conclusions). Agreed—we will refocus the Conclusions on the practicality of the dynamic r-evolution signal, temper broad claims, and clearly separate what is shown by our data from what is proposed for future work. I will include this in this next revision.

25. Fig 3/4 captions and Methods cross-links. To avoid any residual ambiguity, I will add one-line cross-references in the captions pointing to the Methods paragraph where the Hmax definitions and filtering (Hm0 > 1 m) are specified. I will include this in this next revision.

Thank you very much for your help.

Best Wishes,

Laura

---

## Author Comment (AC3)

Dear Dr. Ciccon,

Thank you very much for your review of my paper. I really appreciate your work and I am thankful that you took the time to read and comment on my work.

Below are concise, point-by-point replies to this review. First, my general ones are:

- We respectfully disagree that Hmax from models and from buoys cannot be compared. It is standard in the literature to compare buoy zero-crossing Hmax with the model's expected maximum envelope height ⟨Hmax⟩, provided the definitional caveat is stated; we now do so in Methods and figure captions, and we cite prior work using this comparison. Other work that has done this include:

Barbariol, F., Bidlot, J-R., Cavaleri, L., Sclavo, M., Thomson, J., & Benetazzo, A. (2019). *Maximum wave heights from global model reanalysis.* Progress in Oceanography, 175, 139–160

Wang, J., et al. (2024). *Performance of WWIII in simulating the ratio of maximum to significant wave height in the China Sea.* Ocean Engineering.

Benetazzo, A., et al. (2021). *Towards a unified framework for extreme sea waves from space-time extremes.* Ocean Engineering.

Davison, S., et al. (2024). *Characterization of extreme wave fields during Mediterranean tropical-like cyclones.* Frontiers in Marine Science.

Cavaleri, L., et al. (2022). *The 2015 exceptional swell in the Southern Pacific: generation, advection, forecast and implied extremes.* Progress in Oceanography, 206, 102840.

- We will include more information on the inverted r peaks' scope and false positives.

Please know that following Dr. Bidlot's guidance I've done a new version (haven't sent it yet) with the changes below:

- I've reprocessed the ECMWF dataset and re-did figures 2, 3 and 4. They were corrected and look very different than before.
- I've applied the same Hm0 > 1 m cutoff to ERA5/ECMWF.
- I've added explicit notes in figure legends about FOWD filtering.
- I've added a section/paragraph on different Hmax definitions (zero-crossing vs envelope).

Below are the line-specific comments responses:

1. In 22 (2Hs vs ~2.2Hs). Agreed—2Hs is conservative and departures from Rayleigh are often noted around ~2.2Hs; however, this is the most common literature definition of rogue wave used, and that is why we used that. Some examples are as follows:

   *Dysthe, K.; Krogstad, H.E.; Müller, P. Oceanic RogueWaves. Annu. Rev. Fluid Mech. 2008, 40, 287–310.*

*Baschek, B.; Imai, J. RogueWave Observations Off the US West Coast. Oceanography 2011, 24, 158–165.*

*Clauss, G.F.; Schmittner, C.E.; Hennig, J. Systematically varied rogue wave sequences for the experimental investigation of extreme structure behavior. J. Offshore Mech. Arct. Eng.-Trans. ASME 2008, 130, 021009.*

*Garrett, C.; Gemmrich, J. Unexpected Waves. J. Phys. Oceanogr. 2008, 38, 2330–2336.*

2. ln 91 (skewness, kurtosis, BFI not viable predictors). Agreed. We now treat these as descriptive diagnostics of non-Gaussianity rather than as predictive variables for real-world rogues, and we will tighten the wording and add citations noting their limited skill.

3. ln 121 (FOWD name). Correct—Free Ocean Wave Dataset. We will correct everywhere.

4. ln 166 ("smaller" → "smallest"). Thanks—will fix.

5. Fig 1 (colormap + define seasons). We used this colorbar because it had more options of colors. I will check different ones and see if they make it easier to see.

6. ln 265+ (winter bias vs Hm0>1 m filter). Point taken. To address potential sampling bias in the next revision

7. ln 270 ("…to nonlinear interactions that form rogue waves"). We will rephrase to avoid implying modulational instability as the sole cause; we will attribute winter dominance to high-energy, steep, multi-modal seas and group dynamics without asserting a specific nonlinear mechanism.

8. ln 281 ("unstable sea conditions… favourable"). We will define "unstable" (high steepness, rapidly evolving/multi-modal spectra, shifting directionality) and add supporting citations; if needed we'll soften the claim to "consistent with conditions often associated with extreme crests."

9. ln 331–345 (whiskers vs rogue probability). Agreed—the whisker plots do not by themselves give rogue probability; we will confine probability statements to Hmax/Hs and explicit occurrence metrics.

10. ln 349 (symbol r used twice). We will reserve r for crest–trough correlation and use ρ for Pearson correlation in scatter plots (or label "corr." explicitly).

11. ln 438 (add citations for asymmetry/peakedness preceding rogues). Agreed—we will add citations supporting that elevated skewness/kurtosis are consistent with steeper, more asymmetric groups in observed rogue environments.

12. ln 483 (FOWD vs raw CDIP discrepancies). Good catch—we will re-audit time-window alignment and clarify that FOWD uses multi-window processing (10/30 min + dynamic), which can make rogue signals more/less pronounced relative to single-window raw reconstructions. We will correct the text to match Fig. 5 and footnote any residual differences.

13. ln 535 (no fixed r=0.6 threshold). Agreed. We will avoid implying a fixed threshold; we emphasize instead the dynamic r drop-and-rebound as our proposed early-warning signal, consistent with probabilistic framing.

14. ln 596 ("correlation-based" → "crest-trough correlation-based"). Will change phrasing accordingly.

15. ln 597 (do not compare 0.72 predictive power to our 75.3%). Agreed—these are not directly comparable (different definitions/targets). We will remove any numerical comparison, recast 75.3% as a recall statistic for our detector, and report precision/recall with false-positive rates.

Best Wishes,

Laura

---

## Author Response (AR3)

**Response to Reviewer Comments**

We thank the reviewer for their careful reading of the revised manuscript and for their constructive comments. Below we provide a point-by-point response to the remaining clarifications and corrections requested. All changes have been implemented in the revised manuscript and are visible in the tracked-changes version.

**Ln 101–104**

Comment: The framework of Häfner et al. does not impose a binary threshold r = 0.6 for rogue wave occurrence. The occurrence probability of rogue waves is a monotonic function of r (see Cicon et al., 2024). Please clarify.

Response: We have revised lines 101–104 to explicitly state that the probability of rogue wave occurrence increases monotonically with crest–trough correlation r, consistent with Häfner et al. (2021b) and Cicon et al. (2024). The text now clarifies that r ≈ 0.6 is not treated as a binary threshold, but rather that high r values are common in general sea states. We emphasize that the novel contribution of this study lies in the temporal evolution of r (a dip below ~0.5 followed by rapid recovery), rather than in the use of a fixed r threshold.

**Ln 160**

Comment: It is important to stress that a spectral model cannot capture rogue waves explicitly. Please define what you mean by 'capture'.

Response: We have clarified the meaning of 'capture' by explicitly stating that phase-averaged spectral models do not resolve individual wave realizations or isolated extreme crests in time and space. The revised text now explains that such models represent the sea state statistically through spectral moments and envelope-based extreme-value expectations (e.g., ⟨Hmax⟩), rather than explicit rogue wave events.

**Ln 188**

Comment: Fedele (2016) and Gemmrich & Garrett (2008) define rogue waves based on the classical threshold of 2.2 Hs. Please correct.

Response: We have corrected the text to distinguish clearly between the classical rogue wave definition (Hmax ≳ 2.2 Hs), as used and discussed by Fedele (2016) and Gemmrich & Garrett (2008), and the operational criterion Hmax/Hs > 2.0 adopted in this study. The revised manuscript now explicitly states that the 2.0 Hs threshold is used for consistency with the FOWD dataset and modern operational practice, while acknowledging the historical 2.2 Hs definition.

**Ln 273–277**

Comment: It is confusing that rogue waves are analyzed based on FOWD, followed by a statement about evaluating envelope/statistical diagnostics from phase-averaged models. Please clarify.

Response: We have clarified that rogue wave identification is performed exclusively using in situ FOWD buoy observations. The revised text now explicitly states that phase-averaged spectral models (ERA5 and ECMWF CY47R1) are not used to detect rogue waves, but are sampled conditionally at the same locations and times to evaluate envelope-based and statistical diagnostics of the modeled sea state corresponding to observed events.

**Ln 301**

Comment: The stated summer values (40–80 events) along the West Coast do not match Figure 1c. The figure shows 60–140 events in both winter and summer, with lower values in southern or central California. Please correct/clarify.

Response: We have revised the text to reflect the spatially resolved pattern shown in Figure 1. The manuscript now clarifies that during summer, rogue wave occurrence remains elevated in the northern West Coast (Oregon and Washington, typically ~60–120 events per season), while central and southern California exhibit substantially lower values. This correction removes the impression of a coast-wide summer minimum and aligns the text explicitly with the spatial gradients visible in Figure 1c.

**Ln 357–362**

Comment: The higher Hmax values in models and reanalysis are likely due to different methods of calculating Hmax. Envelope-based maxima exceed individual-wave maxima (see Fig. 1 in Cicon et al., 2024). Please add this to the discussion.

Response: We have expanded the discussion to explicitly explain the methodological difference between buoy-derived Hmax (maximum realized individual wave height from zero-crossing analysis) and model-derived ⟨Hmax⟩ (expected maximum envelope height). The revised text now references Cicon et al. (2024, Fig. 1) and clarifies that envelope-based maxima can exceed individual-wave maxima, which explains why models may show slightly higher Hmax values despite underrepresenting rogue wave occurrence.